# Alternation of Organ-Specific Exposure in LPS-Induced Pneumonia Mice after the Inhalation of Tetrandrine Is Governed by Metabolizing Enzyme Suppression and Lysosomal Trapping

**DOI:** 10.3390/ijms232112948

**Published:** 2022-10-26

**Authors:** Furun Wang, Xue Jiang, Zengxu Yang, Shuang Fu, Shi Yao, Lingchao Wang, Yue Lv, Wenpeng Zhang, Rigao Ding, Xiaomei Zhuang

**Affiliations:** State Key Laboratory of Toxicology and Medical Countermeasures, Beijing Institute of Pharmacology and Toxicology, Beijing 100850, China

**Keywords:** tetrandrine, inhalation, LPS, pulmonary inflammation, lung exposure, lysosomal trapping, CYP suppression

## Abstract

The objective of the present study was to define whether inhaled tetrandrine (TET) could be a promising way to achieve the local effect on its therapeutic efficacy based on biodistribution features using the LPS-treated acute lung injury (ALI) model. The tissue distribution profiles of inhaled TET in normal and ALI mouse models showed that pulmonary inflammation led to an altered distribution in a tissue-specific way. More TET accumulated in almost all tissues including in the blood. Among them, the increased exposure in the lungs was significantly higher than in the other tissues. However, there was a negative increase in the brain. In vitro turnover rates of TET in mouse liver microsomes (MLM) from normal and LPS-treated mice showed significant differences. In the presence of NADPH, TET demonstrated relatively low hepatic clearance (89 mL/h/kg) in that of normal MLM (140 mL/h/kg). Intracellular uptakes of TET in A549, HepG2, RAW264.7, and C8-D1A cells were significantly inhibited by monensin, indicating that the intracellular accumulation of TET is driven by lysosomal trapping. However, in the presence of LPS, only the lysosomal pH partitioning of TET in A549 cell lines increased (~30%). Bidirectional transport of TET across LLC-PK1 cell expressing MDR1 showed that MDR1 is responsible for the low brain exposure via effluxion (ER = 32.46). From the observed overall agreement between the in vitro and in vivo results, we concluded that the downregulation of the CYP3A together with strengthened pulmometry lysosomal trapping magnified the retention of inhaled TET in the lung. These results therefore open the possibility of prolonging the duration of the local anti-inflammation effect against respiratory disorders.

## 1. Introduction

Tetrandrine (TET) is a bisbenzylisoquinoline alkaloid originally isolated from the perennial vine plant *Stephania tetrandra S Moore*, a traditional Chinese medicine (TCM). It has been applied to treat silicosis in China for decades [1,2]. Considerable studies have proved that this compound exhibits a broad spectrum of biological activities, including anti-inflammatory, anti-cancer, immunosuppressant, anti-fibrotic, anti-rheumatoid arthritis, etc. [3,4,5]. Recent studies found that TET is able to inhibit the infections of various viruses, including dengue virus, Ebola virus, Middle East respiratory syndrome coronavirus, and human coronavirus strain OC43 (HCoV-OC43) via the blocking of two-pore iron channels (TPCs), which are located in the membranes of the host endo-lysosome which is the way for viral entry [6,7,8]. Considering respiratory tract infections are often associated with inflammatory response and pulmonary fibrosis, we hypothesized that inhaled TET may be a promising way to achieve the local effects on the treatment of respiratory viral infections even including COVID-19.

Although inhaled drugs offer several benefits for the management of pulmonary diseases over other conventional routes [9,10,11], the development of therapeutic activity is heavily dependent on its pharmacokinetic (PK) behavior on the local target. Therefore, pulmonary PK profiles provide direct information regarding the actual levels of free and pharmacologically active drugs in the lung tissue.

On the other hand, an increasing number of studies have shown that during inflammation and infection, different PK behaviors will occur mainly due to the alternations of drug-metabolizing enzymes, transporters, biomembrane permeability, etc. [12,13,14,15]. Thus, the characterization of drug pharmacokinetics in healthy human subjects often fails to adequately describe the dose–exposure response relationships occurring in the targeted patient population. The time of disease onset and the duration of infection, inflammation, or concomitant stressors remain unaddressed questions [16,17,18]. Importantly, disease-induced PK changes may ultimately lead to changes in efficacy and toxicity. However, with the current methodology, obtaining these data in clinical patients is difficult. More studies are needed to improve our ability to predict the impact of disease on drug disposition and associated outcomes. Comparing the characteristics of PK and tissue distribution between normal animals and disease animal models is a good way to reflect the potential pharmacokinetic alternation in the disease state.

Although a number of investigations into the characteristics of disease-PK and inhalation PK have been described in recent years [13,19,20,21,22,23], to the best of our knowledge to date, there are no acquired systemic PK and biodistribution profiles of an inhaled drug against respiratory viral infections under an acute pneumonia state. An evaluation of such kind would provide the required confidence in clinical transformation for inhalation drugs.

The objective of the present study was to compare the PK and tissue distribution characteristics of TET following inhalation in normal and pneumonia mice based on a validated tracheal administration lipopolysaccharide (LPS)-induced acute pneumonia mouse model in the first step. Furthermore, as a basic lipophilic drug, the mechanisms responsible for the biodistribution of TET under an inflammatory state were closely paid attention to. The role of metabolizing enzymes and lysosomal trapping on the TET disposition in an inflammation state was addressed using in vivo and in vitro approaches. It is hoped that this work will contribute to a deep understanding of the biodistribution of TET in acute inflammatory states of the respiratory system.

## 2. Results

### 2.1. Features of the ALI Mouse Model Induced by LPS

The international scoring standard was applied to identify the histological evidence of lung injury [24]. Compared to the normal saline (NS) control group (no injury), the LPS challenge exhibited obvious acute lung injury (ALI). Maximal lung injury appeared within 24 h of LPS exposure (Figure 1A). The inflammatory response was measured simultaneously. Maximum proinflammatory cytokines (IL-6 and TNF-α) levels were reached 6–12 h after LPS treatment (Figure 1B), which agreed with the histological damage that occurred in the lungs. Collectively, 10 mg/kg LPS 6h-pretreated via the intratracheal (i.t.) route successfully produced an ALI mouse model. This was the first step toward further investigation.

### 2.2. Effect of ALI on TET PK Behavior and Biodistribution

Pharmacokinetic parameters of TET in blood after a single i.v. administration in normal and LPS-induced ALI mice are presented in Table 1. After i.v. injection at the same dosage, initial TET concentrations in the blood were almost identical. However, the average AUC in LPS-treated mice yielded a significant increase (~1.6–2.1-fold, *p* < 0.01) compared to the control group. This change was associated with strongly reduced systemic clearance (~47%, *p* < 0.01) combined with increased V_z_ (~134%, *p* < 0.05), while the V_ss_ did not exhibit an obvious difference.

The observed concentration–time profiles and corresponding pharmacokinetic parameters for TET via i.t. administration in different tissues and blood obtained from control and LPS-treated mice are shown in Figure 2 and Table 2. The average AUC and K_p_ of TET in different tissues in normal and LPS-treated mice are visualized in Figure 3. It was found that when compared to the saline group, the systemic blood exposures of TET via the i.t. route was increased in the LPS group as well (~1.25-fold, *p* < 0.05). TET displayed dramatic tissue accumulation after i.t. administration both in normal and LPS-treated mice. In normal mice, the biodistribution of TET seemed to be divided into three groups. The first group included the spleen (with K_p_ of 45.7), followed by the kidney (K_p_ of 37.5), liver (K_p_ of 28.0), and lung (K_p_ of 24.7), which showed the highest accumulation of TET. The intestine, stomach, and heart produced a relatively higher accumulation of TET with K_p_ ranging from 8.3 to 5.1. The remaining tissues, i.e., adipose, muscle, and brain had a relatively lower TET burden with K_p_ ranging from 2.4 to 1.0. Furthermore, in the LPS-treated group, a widespread increase in TET accumulation in tissues throughout the body occurred. Among those, there was a remarkable elevation in the lung. Apart from this, the biodistribution pattern of TET did not alter significantly.

### 2.3. Impact of Inflammation on CYP-Associated TET Metabolism

To ascribe the responsibility of CYP for the clearance of TET and to ascertain whether this was linked to inflammation, in vitro turnover rates of TET and specific CYP3a probe substrates (midazolam and testosterone) in mouse liver microsomes (MLM) and pulmonary microsomes (MPM) from normal and LPS-treated mice were compared. The outcomes are presented in Figure 4 and Table 3. The influence of pulmonary inflammation on the activities of CYP3a was measured by calculating the formation velocities of 1′-OH- midazolam and 6-β-OH- testosterone. Firstly, it was noted that the activity of CYP3a from the liver was significantly stronger than that from the lung carrying almost three-order magnitude differences. The contribution of the hepatic metabolism was overwhelming dominant. Next, even regarding CYP3a, the impact of inflammation on the metabolism of the different substrate was different. As for midazolam, the 1′-hydroxylation reaction was obviously lower (~2.55-fold) by LPS-induced pneumonia at 24 h (Figure 4A, *p* < 0.01). By contrast, no obvious alternation was observed for the testosterone 6-β-hydroxylation reaction under an inflammation state (~1.14-fold, Figure 4A). The turnover rates of TET in normal and LPS-treated MLM (Figure 4B) displayed significant differences. In the presence of NADPH, TET demonstrated a relatively low hepatic clearance (140 mL/h/kg) in normal MLM. However, the hepatic clearance (89.0 mL/h/kg) was greatly reduced in MLM following inflammatory stimulation.

### 2.4. The Role of Lysosomes in the TET Cellular Distribution and the Effects of Inflammation

The cellular uptake of TET in various organs was determined using A549, HepG2, RAW264.7, and C8-D1A cells in the presence or absence of a lysosomal inhibitor (monensin). The effect of inflammation on the lysosomal trapping of TET was also investigated in LPS-treated cell models. RAW264.7 macrophage is a classic model in inflammation research. The elevation of IL-6 in the culture medium of all cell lines reflected the success of the in vitro inflammation model (Figure 5A). The absolute cellular uptakes of TET (Figure 5B,C) in different groups manifested two characteristics: (1) deactivation of lysosomes by monensin (25 µM [25,26]) greatly reduced the intracellular concentrations of TET in normal and LPS-treated cells; (2) in the inflammatory state, intracellular TET concentrations in A549 cell were significantly increased (121.7~132.1%). However, the same trend was not observed in other cells.

### 2.5. Roles of Drug Transporters Involved in TET Cellular Accumulation

The transcellular permeability of TET across monolayer and transporter-mediated active transportation was evaluated using transfected cell lines expressing OATP1B1, OATP1B3, or MDR1 protein. The corresponding results are presented in Table 4 and Table 5. The uptake of TET into OATP1B1 and OATP1B3 expressing HEK293 cells was not significantly higher than that into vector-transfected mock cells (uptake ratio of ~1.2). In parallel, we further confirmed that OATP1B1 and OATP1B3 were not involved in the cellular uptake of TET by the addition of a specific inhibitor, while the uptake of selective reference compound (estradiol-17β-D-glucuronide) mediated by OATP1B1 and OATP1B3 was positive.

Bidirectional transport data acquired during the LLC-PK1 experiments using mock cells and MDR1-transfected cells showed that MDR1 was responsible for the low permeability of TET via effluxion. Moreover, the incomplete inhibition of TET efflux after the combination of a P-gp inhibitor (~51%) suggested the possible involvement of other efflux transporters in addition to P-gp.

## 3. Discussion

Respiratory system disorders affect millions of people worldwide every year [27,28]. The development of potent, targeted treatments to reduce the inflammatory reactions in the lung and re-establish homeostasis without harming other organs is still an unmet demand [29,30]. Tetrandrine has been used in China for more than half a century for the treatment of pulmonary inflammation-related diseases with solid evidence of clinical efficacy [3,4,31]. However, systemic pharmacokinetic studies on TET remain incomplete. The pharmacokinetic and biodistribution profiles of TET in disease states are even less well documented. A deep investigation of the exposure characteristics and the associated mechanisms in in vivo and in vitro animal models can offer critical information for the development of clinical escalation of TET.

The development of in vivo and in vitro acute pulmonary inflammation models is the basis for investigating the altered pharmacokinetic behavior of TET. Although it is recognized that extrapolation from animal data to humans is difficult due to the potential of species differences, rodent models remain the best way to assess the impact of inflammation triggered by LPS on drug response and disposition [32]. Numerous reports on treated rodents have shown a significant increase in drug exposure during inflammation [33]. In order to reproduce acute lung infections in mice, LPS was dosed via airway administration in our study. The characterized inflammatory response that involved a rapid increase in major proinflammatory cytokines release and histological damage in the lungs indicated the development of ALI in the mice model (Figure 1). For a thorough exploration of the intracellular accumulation of TET in different organs under inflammatory stimulation, in vitro cell models using four types of tumoral cells exposed to LPS were established. All these cells were able to respond to an inflammatory stimulus after determining the secretion of IL-6 in the cell culture medium (Figure 5A).

TET was widely distributed in tissues and organs throughout the body after i.t. administration in normal mice. It was noted that the concentrations were significantly higher in tissues rich in lysosomes (spleen, kidney, liver, and lung) [26,34]. As a basic lipophilic compound, it is highly possible to be sequestrated in these tissues through pH-driven lysosomal trapping, which could produce enormous concentrations within the lysosomal subcompartment. Whether tissue accumulation of TET is affected by augmenting of lysosomal trapping in response to inflammation drove us to conduct a tissue distribution study in the same inhaled manner in an ALI mice model. Parallel tissue distribution profiles showed that in inflamed mice, the distribution of inhaled TET in almost all organs exceeded that of normal mice (Figure 2). Of these, changes in the lungs were the most eye-catching with a remarkable increase in lung retention and organ tissue-to-blood partition coefficient (K_p_) (from 24.7 to 45.8). We also noted that in addition to increased TET exposure in lysosome-rich tissues, there was an enhancement in the circulating system (~1.25 fold compared to the normal group). The comprehensive analysis suggests that disease-PK is highly complex where multiple factors may be involved.

To exclude complicating factors, we compared the pharmacokinetic behaviors of normal mice and ALI model mice after i.v. administration of TET at 5 mg/kg. The consequences of pharmacokinetic parameters showed a 1.68-fold increase in blood AUC, a decrease in clearance to 47%, and a mild 1.36-fold increase in the steady-state volume of distribution (V_ss_) in inflamed mice. This suggests that inflammation may primarily attenuate TET metabolism and clearance, accompanied by an elevated tissue distribution, which is generally in accordance with the outcomes of biodistribution. Therefore, several additional in vitro experiments were designed to ascribe the impacts of inflammation on TET metabolic clearance and lysosomal trapping. CYP3A is one of the most important cytochrome P450 isoforms responsible for drug metabolism. According to our previous results that showed that CYP3A is the major isoform toward TET (f_m_ > 90%), the metabolic rates of TET, as well as midazolam and testosterone, were compared in liver and lung microsomes prepared from normal and inflamed mice. It is interesting to note that despite being the same substrate of CYP3A, the effects of inflammation on the metabolism of midazolam and testosterone by CYP3A were markedly different. In the presence of acute inflammation, the metabolic turnover of TET in mouse liver microsomes demonstrated a significant slowdown with a decreased estimated CL_hep_ (~63.6%, *p* < 0.01), which was close to the in vivo outcomes (~47%, *p* < 0.01). Regarding lung-resourced metabolism, both the probe drugs of CYP3A and TET appeared to have lower metabolic transformations. Thus, the observed overall agreement between the in vitro and in vivo metabolic clearances in response to inflammation reflected the contribution of the hepatic metabolizing enzyme. A significant amount of evidence has been reported on the relationships between inflammation on drug-metabolizing enzyme activity [13,21,35]. However, discrepancies still exist in available investigations. Martinez et al. [21] suggested that some of the apparent dissimilarities may be attributed to animal species and the duration of exposure to these cytokines.

Apart from the inflammation-associated regulation of CYP and thus TET elimination, changes in the volume of distribution and tissue biodistribution profiles suggested that inflammation is able to affect the intracellular espouse of TET. With the ability to transport protons across cellular lipid membranes, monensin can raise the lysosomal pH and reduce the difference in pH between lysosomes and the surrounding cytosol [36]. Intracellular uptake in the current four cell lines was significantly inhibited by monensin, indicating that intracellular accumulation of TET is driven by lysosomal trapping. Considering that many weakly alkaline drugs are lysosomotropic compounds when a cell is exposed simultaneously to two different lysosomotropic compounds, they compete for intra-lysosomal protonation and their relative lysosomal accumulation depends on their relative concentrations and their respective pK_α_ values. However, in the presence of LPS, only the lysosomal pH partitioning of TET in A549 cell lines increased (~30%) (Figure 5B). With monensin present in the LPS-treated incubation, the uptake in cells was still inhibited to a similar degree (~40%).

In acutely inflamed mice, a larger magnitude of increase in tissue accumulation and distribution after TET inhalation administration appeared in the lungs with the AUC ratio of 2.32, which was higher than the increase in blood AUC (1.25). Together with in vitro cells and in vivo tissues, it is clear that that during inflammation, inhaled TET was exposed significantly more in the lungs than in other tissues due to increased organ-specific lysosomal trapping and attenuated hepatic metabolism.

Another notable phenomenon was that inhaled TET was significantly less abundant in the brain than in other tissues in normal mice and was the only tissue to undergo a reduction in inflamed mice. LLC-PK1 cell expressing MDR1 was applied to estimate the potential role of efflux transporter for TET across the BBB. The results of the bidirectional cellular transport displayed that P-gp did involve in the efflux transport of TET with positive permeability obviously reduced and efflux significantly increased (Table 5) in transfected cells. This could explain the low brain exposure of TET in normal mice. The impact of inflammation on P-gp has been well investigated for several years [15]. However, to date, in vitro studies using different models of BBB have provided contradictory results on P-gp after the exposure of cells to cytokines. A downregulation of P-gp was observed after exposure of isolated rat brain capillaries to TNF-α, whereas increased P-gp expression and activity were found following a more prolonged exposure [37,38]. In the current case, even when inflammation led to a diminished TET metabolism and no alternation of lysosomal trapping in astrocytes, the reduced level in the brain may have been caused mainly by the enhanced activity of P-gp. The exact reason requires further investigation.

## 4. Materials and Methods

### 4.1. Chemicals and Reagents

TET (purity > 99.2%) was obtained from Conba Pharmaceutical Co., Ltd. (Hangzhou, China). Buspirone hydrochloride (purity > 99%, internal standard), 17β-estradiol-17β-D-glucuronide, rifampicin, lipopolysaccharide, and substrates and metabolites of CYP were bought from Sigma-Aldrich (St. Louis, MO, USA). DMSO (purity > 99.7%) was purchased from Innochem (Beijing, China). The HPLC-grade acetonitrile (ACN) and Pierce BCA protein assay kit were purchased from Thermo Fisher Scientific (Waltham, MA, USA). The HPLC-grade formic acid was bought from J&K Scientific (Beijing, China). Monensin was bought from TCI Shanghai (Shanghai, China). Tariquidar (TAR) was purchased from Shanghai Yuanye Bio-Technology Co., Ltd. (Shanghai, China). Atenolol and digoxin were purchased from Selleck Chemicals (Houston, TX, USA) and metoprolol was purchased from Macklin (Shanghai, China). The RM-003 liquid atomizer was purchased from Raymain Information Technology Co., Ltd. (Shanghai, China). IL-6 and TNF-α ELISA kits were purchased from Mlbio (Shanghai, China).

### 4.2. Animals

Healthy male BALB/c mice (19–23 g) were purchased from Beijing Vital River Laboratory Animal Technology Co., Ltd. The animals were kept in a clean facility with a 12-h light/dark cycle and fed chow and water ad libitum at 22 °C and 50% relative humidity. All the experiments were conducted at Beijing Center for Drug Safety Evaluation after obtaining ethical approval (IACUC-DWZX-2021-662) from the Institutional Animal Care and Use Committee of the Centre, which followed the guidelines of the Association for Assessment and Accreditation of Laboratory Animal Care International (AAALAC).

### 4.3. Construction of the LPS-Induced Mouse Model of ALI

Fifteen male BALB/c mice were randomly divided into five groups (*n* = 3 for each group), including one control group and four model groups. After anaesthetization with aerosolized isoflurane, model mice were subjected to intratracheal (i.t.) administration of 50 μL LPS (10 mg/kg, 5 mg/mL dissolved in 0.9% normal saline (NS)) through an RM-003 liquid atomizer after adequate exposure of the trachea with a laryngoscope. Control mice were administrated in a similar manner with the same volume of only 0.9% NS. Control groups were sacrificed at 24 h post-dosing NS, and model groups were sacrificed at 6, 12, 24, and 144 h after dosing, respectively. The right lungs were harvested for the determination of inflammatory cytokines, and the left lungs were fixed with 4% formaldehyde for HE staining and imaged on a Leica Microsystem. The images (at least five random 20× fields per lung) were scored in accordance with the international scoring standard to calculate the total score of the lung injury [24]. IL-6 and TNF-α of lung homogenate were tested with ELISA kits according to the protocol. A schematic diagram of this experimental procedure is presented in Figure 6A.

### 4.4. Cell Culturse

A549 cells, HepG2 cells, C8-D1A cells, and RAW264.7 were obtained from BeNa Culture Collection (Kunshan, China). The recombinant HEK293 cell lines expressing human OATP1B1 (HEK293-OATP1B1), OATP1B3 (HEK293-OATP1B3), and MOCK transfected with empty vector (HEK293-MOCK) were supplied by GenoMembrane (Yokohama, Japan). LLC-PK1 cell with stable expressing human MDR1 and mock-transfected with empty vector (LLC-PK1-MOCK) were gifted by the lab of Professor Qingcheng Mao (University of Washington, Seattle, WA, USA). All cells were cultured in 1640 medium supplemented with 10% fetal bovine serum (FBS), and streptomycin–penicillin (100 U/mL), at 37 °C in a humidified 5% CO_2_ incubator.

### 4.5. In Vivo Experiments

#### 4.5.1. Pharmacokinetics after i.v. Administration

Twelve male BALB/c mice were randomly divided into two groups (*n* = 6 for each group). The control group received a single i.v. administration of 5 mg/kg TET, and the LPS group were i.v. administrated with 5 mg/kg TET 6-h after i.t. injection of 50 μL LPS (5mg/mL) to produce acute pneumonia. The whole blood samples were collected at 0.033, 0.083, 0.25, 0.5, 1, 2, 4, 8, 24, 48, 72, 96, 120, and 144 h after TET administration and stored at −40 °C for LC-MS/MS analysis.

#### 4.5.2. Tissue Distribution after i.t. Delivery

The different tissue distribution characteristics of TET in normal and pneumonia model mice via i.t. delivery were investigated. Thirty BALB/c mice were randomly divided into ten groups (*n* = 3 for each group). Five groups of mice were intratracheally injected with 15 mg/kg TET (dissolved in 22.5 mg/mL aspartic acid solution) as control groups, and the other five groups received i.t. administration of 15 mg/kg TET 6 h after i.t. injection of 50 μL LPS (5 mg/mL) as model groups. One control group and one model group were sacrificed simultaneously at the set time points of 2, 4, 8, 24, and 144 h after the administration of TET. After a cardiac puncture to obtain blood samples, tissue samples including heart, liver, spleen, lung, kidney, stomach, intestine, and brain were immediately harvested and all the tissues were thoroughly rinsed in ice physiological saline. All the tissue samples were weighed and stored with blood samples at −40 °C for LC-MS/MS analysis.

### 4.6. In Vitro Experiments

#### 4.6.1. Preparation of Mouse Liver Microsomes and Pulmonary Microsomes

Liver and pulmonary microsomes from normal and LPS-induced pulmonary infected mice were isolated according to the conventional differential centrifugation approach [39]. In brief, six normal mice and six acute pneumonia mice (48 h after i.t. administration of LPS) were sacrificed. Lungs and livers were harvested and rinsed with ice NS. The tissues were weighed and homogenized with 50 mM Tris buffer solution (TBS, m:v = 1:3). The prepared homogenate was then immediately centrifuged at 10,000× *g* at 4 °C for 20 min. The resulting supernatant was further centrifuged at 105,000× *g* at 4 °C for 60 min to yield mouse liver microsomes (MLM) and mouse pulmonary microsomes (MPM). The sediment was resuspended with storage buffer (50 mM TBS containing 0.25 M sucrose and 20% glycerol) and stored at −80 °C immediately. The protein content of the harvested microsomes was measured following the BCA protein assay protocol.

#### 4.6.2. Metabolic Stability of TET in MLM and MPM from Normal and Acute Pneumonia Mice

TET (1 μM, 1% acetonitrile) was incubated in normal or pneumonia mice liver microsomes or pulmonary microsomes (0.5 mg/mL) diluted with 100 mM PBS (pH 7.4) containing MgCl_2_ (3 mM). After preincubation of 5 min, the reaction was initiated by adding NADPH (final concentration of 1 mM) at 37 °C. Periodic aliquots of the incubation mixture at 0, 15, 30, and 45 min were removed, and the protein was precipitated with ice-cold acetonitrile containing IS. After centrifugation, the supernatant was collected and analyzed for the depletion of TET by LC-MS/MS. Negative control incubations in the absence of NADPH and positive control incubations in the presence of cocktailed CYP probes including midazolam (2.5 μM) and testosterone (5 μM) were conducted simultaneously.

#### 4.6.3. Quantification of Lysosomal TET Content in Normal and LPS-Induced Cell Lines

A549, HepG2, RAW264.7, and C8-D1A cells were seeded in 24-well culture plates at a density of 1 × 10^5^ cells/well and cultured for 36 h to form an 80–90% confluent monolayer for the subsequent experiments. The lysosomal accumulations of TET in different cell lines with or without the impact of LPS-induced inflammation were measured in control and LPS-pretreated cells containing 2 μM TET only, and in control and LPS-pretreated cells joint addition of 25 µM monensin and 2 μM TET to inactivate lysosomes, respectively. The schematic diagram of the experimental procedure is shown in Figure 6B. All the cells were cultured as described previously. After two rounds of preheated PBS washing, cells were incubated in 200 μL of loading solution for 40 min at 37 °C with 5% CO_2_. At the end of the incubation, the medium was collected for the measurement of IL-6. The cells were washed three times with ice-cold Hank’s Balanced Salt Solution (HBSS). Finally, cells were frozen and thawed for three cycles with 300 μL pure water to obtain cell lysate. Then, cell lysates were collected and stored at −40 °C for LC-MS/MS analysis and the measurement of protein with a Pierce BCA protein assay kit.

#### 4.6.4. Uptake Study of TET by OATP1B1 and OATP1B3 Expressing the HEK293 Cell Line

Expressing OATP1B1/OATP1B3 or MOCK HEK293 cells were utilized to evaluate whether TET is a substrate of these two SLC transporters. Cell culture was performed as mentioned above. Cells were seeded on 96-well plates at the density of 0.8 × 10^5^ cells/well. The culture medium was replaced with a medium containing 5 mM sodium butyrate for 24 h before the transport study to enhance the expression of transporters. For the transport assay, the DMEM media were removed and the cells were washed with pre-warmed Krebs–Henseleit buffer twice (0.1 mL/well). The second washing buffer was kept in the wells and the plate was incubated at 37 °C for 5 min before the uptake assay was initiated. After the washing buffer was removed, the pre-warmed TET (5 μM) or a mixture of TET and selective inhibitor (100 μM rifampicin) was added into each well and the plates were incubated at 37 °C for 5 min. The incubation solution was removed and the cells were washed with ice-cold transportation buffer three times (0.1 mL/well). A total of 0.1 mL pure water was added to each well to lyse the cells by repeated freezing and thawing (−196 °C–37 °C, three times). The cell lysate was collected and stored at −40 °C for LC-MS/MS analysis and protein measurement. The negative control (HEK293-MOCK cells) and positive control (20 μM 17β-estradiol-17β-D-glucuronide) were included simultaneously.

#### 4.6.5. Bidirectional Transport Study of TET by MDR1 Expressing the LLC-PK1 Cell Line

MDR1 expressing LLC-PK1 cells or MOCK LLC-PK1 cells were utilized to identify whether P-gp is involved in the efflux transport across the membrane barrier. Cells were cultured as described above. Cells were seeded at a density of 2 × 10^4^ in Millicell^®^ 24-well cell culture plates with a 6.5 mm insert (0.4 µm polycarbonate membrane) from Corning (Kennebunk, ME, USA) 7 days before the experiments. On the day of the experiment, transepithelial electrical resistance (TEER) was measured to assess monolayer integrity. After washing the cells twice with HBSS, transport experiments in either the apical-to-basolateral direction (A-B) or the basolateral-to-apical direction (B-A) were conducted. The donor solution (HBSS with 1% BSA) containing 2 μM TET or cocktailed positive control drugs (10 μM metoprolol for high permeability, 10 μM atenolol for low permeability, and 2 μM digoxin for P-gp substrate) was added into the apical or basolateral compartment and a sample (50 μL) was immediately taken as C_0_. TAR (5 μM) as a selective MDR1 inhibitor was co-incubated with TET or positive control compounds. After incubating the plates at 37 °C for 2 h, samples from the receiving and donor wells (50 μL) were taken simultaneously. All the samples were stored at −40 °C for LC-MS/MS analysis.

### 4.7. Bioanalytical Methods

All samples of tissue homogenate, plasma, cell lysate, and cell culture medium were precipitated with acetonitrile (containing IS, buspirone, 5 ng/mL) followed by quantitation against the corresponding standard curve prepared in the blank biomatrix. The concentration of TET was determined by an LC-MS/MS method with an API 5000 Triple quadrupole mass spectrometer (AB Sciex, Foster City, CA, USA) connected to a Shimadzu LC-20AD HPLC system (Shimadzu, Japan). The chromatographic column was the C18 column (3.0 mm × 50 mm, 2.6 µm, Phenomenex). The mobile phase consisted of water containing 0.1% of formic acid (A) and acetonitrile containing 0.1% of formic acid (B). Separation was achieved following a binary gradient elution procedure: 0–0.3 min B 10%, 0.3–1.4 min B 10%→90%, 1.4–1.8 min B 90%, 1.8–1.9 min B 90%→10%, 1.9–3.0 min B 10%. The volume of each injection was 5.0 µL, and the flow rate was 0.6 mL/min.

TET and IS were detected by multiple reaction monitoring (MRM) in the positive ion mode. The precursor and product ions used for quantification were as follows: *m*/*z* 609.0→381.1 for TET and *m*/*z* 386.4→122.1 for IS, respectively.

### 4.8. Data Processing and Statistical Analysis

Pharmacokinetic parameters including t_1/2_, AUC_(0-t)_, AUC_(0-∞)_, and MRT were calculated using WinNonlin 7.0 (Pharsight, CA) by the non-compartmental model. Initial plasma concentration post i.v. (C_0-i.v._) was extrapolated from the first three points of the logarithmic plasma concentration. The area under the blood concentration–time curve from t = 0 to infinity (AUC_inf_) was estimated using the linear trapezoidal rule. Blood clearance (CL) was calculated as the i.v. dose divided by AUC_inf_. Apparent steady-state distribution volume (V_ss_) was determined by the clearance multiplied by mean residence time. The distribution volume in the elimination phase (V_z_) was determined by the clearance divided by the first-order rate constant of the terminal elimination phase (λ_z_). The tissue partition ratio (K_p_) of each tissue was calculated as Equation (1).
(1)$$K_{p}={\mathrm{AUC}}_{\left(0-\infty ,\mathrm{tissue}\right)}/{\mathrm{AUC}}_{\left(0-\infty ,\mathrm{plasma}\right)}$$

The in vitro t_1/2_ in mouse liver microsomal incubation was calculated from the semi-log plot of percentage remaining vs. incubation time and intrinsic clearance (CL_int,app_ mL/min/mg protein) was calculated as Equation (2).
(2)$${\;\mathrm{CL}}_{\mathrm{int},\mathrm{app}}=\frac{0.693}{{\mathrm{in}\;\mathrm{vitro}\;t}_{1/2}}\times \frac{\mathrm{volume}\;\mathrm{of}\;\mathrm{incubation}\;\left(\mu l\right)}{\mathrm{amount}\;\mathrm{of}\;\mathrm{microsomal}\;\mathrm{protein}\;\mathrm{in}\;\mathrm{incubation}\;\left(\mathrm{mg}\right)}$$

Next, the scaled intrinsic clearance (CL_int,scaled_) and hepatic blood clearance (CL_h,blood_) were calculated according to Equations (3) and (4), where 90 mL/min/kg and 87.5 g liver/kg body weight were used for hepatic blood flow (Q) and physiological constants for mouse (Davies and Morris, 1993), f_up_, f_u,mic_, and R_b/*p*_ were obtained from measured values.
(3)$${\mathrm{CL}}_{\mathrm{int},\mathrm{scaled}}={\mathrm{CL}}_{\mathrm{int},\mathrm{app}}\times \frac{45\mathrm{mg}\;\mathrm{microsomes}}{g\;\mathrm{liver}}\times \frac{g\;\mathrm{liver}}{\mathrm{kg}\;\mathrm{body}\;\mathrm{weight}}$$
(4)$${\mathrm{CL}}_{h,\mathrm{blood}}=\frac{Q\times f_{u,p/}\mathrm{Rb}/p\times {\mathrm{CL}}_{\mathrm{int}}/f_{u,\mathrm{mic}}}{Q+{\mathrm{CL}}_{\mathrm{int}}\cdot f_{u,p/\mathrm{Rb}}\cdot {\mathrm{CL}}_{\mathrm{int}}/f_{u,\mathrm{mic}}}$$

In the uptake study of TET by OATP1B1 and OATP1B3 expressing the HEK293 cell line, the uptake clearance (U) of the substrate was calculated according to Equation (5), the transporter-mediated uptake ratio (UR) was calculated by dividing the uptake clearance from transporter-expressing cells to that from the mock cells according to Equation (6), and the inhibition ratio (IR) was calculated according to Equation (7), where *p* represents the protein content and T is the duration of the incubation time.
(5)$$U=\frac{C_{\mathrm{lysate}}}{p\times T}$$
(6)$$\mathrm{UR}=\frac{U}{U_{\mathrm{MOCK}}}$$
(7)$$\mathrm{IR}=(1-\frac{U_{\mathrm{with}\;\mathrm{inhibitor}}-U_{\mathrm{mock}\;\mathrm{with}\;\mathrm{inhibitor}}}{U_{\mathrm{without}\;\mathrm{inhibitor}}-U_{\mathrm{mock}\;\mathrm{without}\;\mathrm{inhibitor}}})\times 100$$

In the bidirectional transport study in LLC-PK1 cells, the following equations were used to calculate P_app_ values and the MDR1 efflux ratio.
(8)$$P_{\mathrm{app}}=\frac{1}{A\times C_{0}}\times \frac{d_{x}}{d_{t}}$$
(9)$$\mathrm{ER}=\frac{P_{\mathrm{app},\mathrm{BA}}}{P_{\mathrm{app},\mathrm{AB}}}$$
where A is the surface area of the transwell insert, C_0_ is the initial concentration of a compound applied to the donor chamber, t is incubation time, *x* is the amount of a compound in the receiver compartment, and d_x_/d_t_ is the flux of the compound across the cell monolayer.

The experimental data are shown as the mean ± standard deviation (SD). The pharmacokinetic parameters estimated from WinNonlin are shown as the mean ± standard error (SE). Statistically significant differences between the two groups were determined by the Student’s *t*-test. Differences were considered to be significant at *p* < 0.05.

## 5. Conclusions

From the present study, the characterizations of inhaled TET PK in healthy versus LPS-treated mice were achieved. From the observed overall agreement between the in vitro and in vivo results, we concluded that the downregulation of the CYP3A together with strengthened pulmometry lysosomal trapping magnified the retention of inhaled TET in the lung. These results have opened the possibility of prolonging the duration of local anti-inflammation effect against respiratory disorders. Our work will add evidence to TET pharmacokinetic alternation and related mechanisms during inflammation.

## Figures and Tables

**Figure 1 ijms-23-12948-f001:**
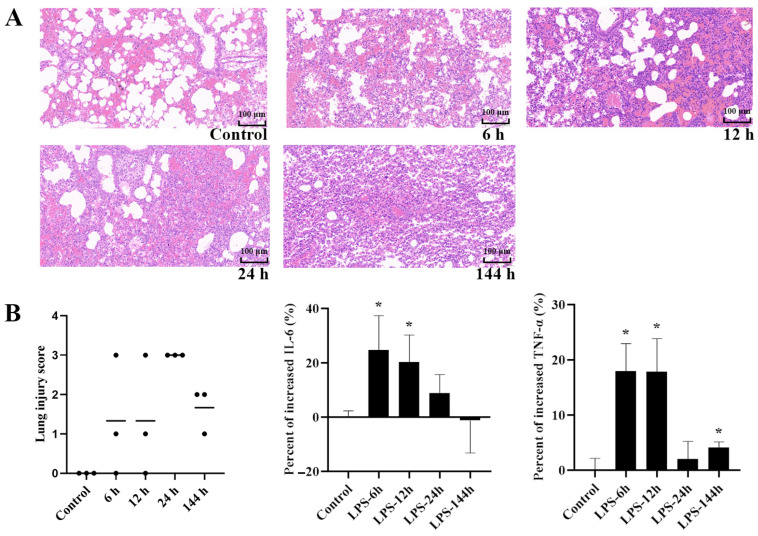
The lung inflammation and injury in LPS-induced ALI mice at different time points. (**A**) Histological image (×20) of lung sections with hematoxylin and eosin (H&E) staining. (**B**) Lung injury scores (from 0 to 4) of the lung sections and the increased levels of IL-6 and TNF-α (*n* = 3, Mean ± SD, * *p* < 0.05 compared to the control group).

**Figure 2 ijms-23-12948-f002:**
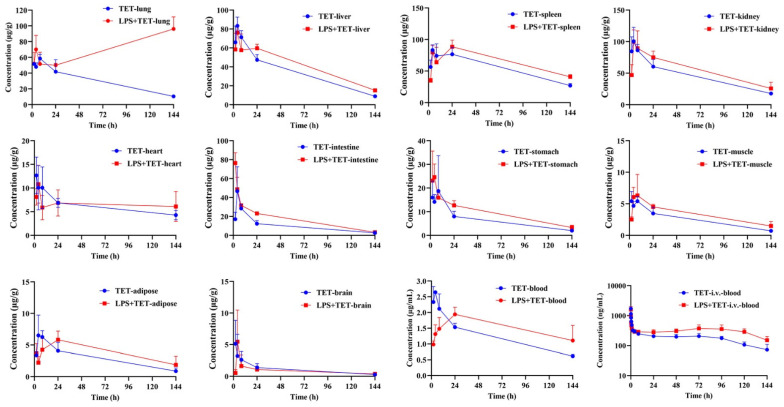
Mean biodistribution versus time profiles of inhaled TET and blood concentration versus time curves via i.v. administration of TET in normal and LPS-induced ALI mice (*n* = 3, Mean ± SD).

**Figure 3 ijms-23-12948-f003:**
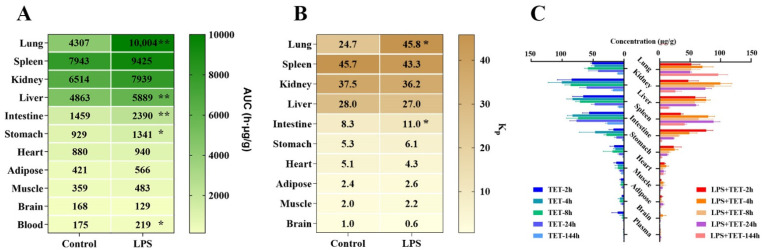
Biodistribution characters of TET in major tissues in normal and LPS-induced mice. Mean AUCs (**A**) and K_p_ (**B**) of TET in different tissues after a single i.t. administration of 15 mg/kg TET in normal and LPS-induced mice (*n* = 3, * *p* < 0.05, ** *p* < 0.01 compared to the normal mice group). (**C**) Comparative biodistributions of TET in major tissues in normal and LPS-induced mice at different time points.

**Figure 4 ijms-23-12948-f004:**
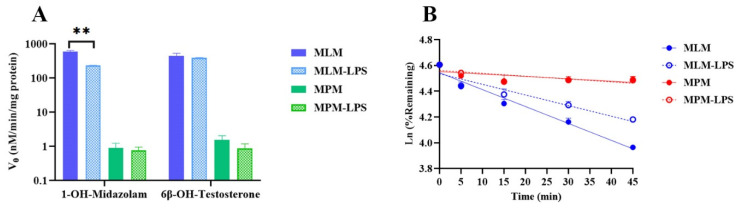
The activity of CYP3A in normal and LPS-induced ALI mice liver and lung microsomes. (**A**) The metabolic transformation velocity of midazolam and testosterone to 1-OH-Midazolam and 6β-OH-Testosterone in normal and LPS-induced ALI mice liver and lung microsomes (*n* = 3, Mean ± SD, ** *p* < 0.01 compared to the normal mice group). (**B**) The metabolic stability of TET in normal and LPS-induced ALI mice liver and lung microsomes (*n* = 3, Mean ± SD).

**Figure 5 ijms-23-12948-f005:**
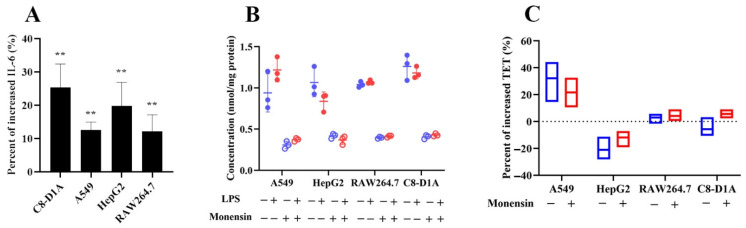
The uptake of TET in normal and LPS-induced cells in the presence or absence of monensin. (**A**) The increased levels of IL-6 after being induced by LPS. (**B**) The uptake of TET in four kinds of cells under different conditions. (**C**) The increased levels of TET in different cells after being induced by LPS (*n* = 3, Mean ± SD, ** *p* < 0.01 compared to the normal cell group).

**Figure 6 ijms-23-12948-f006:**
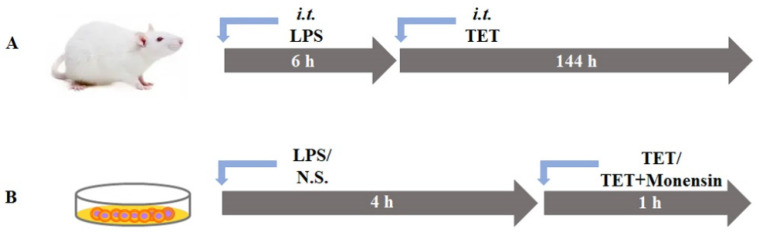
The schematic diagram of the LPS-induced ALI mice model for PK and biodistribution investigation (**A**) and various LPS-treated cell models to investigate cellular accumulation by lysosomal trapping (**B**).

**Table 1 ijms-23-12948-t001:** Pharmacokinetic parameters of TET in blood after a single i.v. administration in normal and LPS-induced ALI mice (*n* = 6, Mean ± SD).

Parameters	Unit	Normal Mice	LPS-Treated Mice
t_1/2_	h	30.7 ± 5.21	93.2 ± 26.4 **
C_0_	ng/mL	2273 ± 270	2300 ± 935
AUC_(0-t)_	h·ng/mL	26253 ± 3417	44098 ± 8457 **
AUC_(0-∞)_	h·ng/mL	29451 ± 3794	65735 ± 17527 **
MRT_(0-t)_	h	72.7 ± 6.40	137 ± 40.4 **
V_ss_	L/kg	7.71 ± 2.01	10.5 ± 1.77
V_z_	L/kg	12.5 ± 1.63	10.3 ± 1.75 *
CL	mL/h/kg	172 ± 22.1	80.9 ± 22.3 **

Twelve male BALB/c mice were randomly divided into two groups (*n* = 6). The control group received a single i.v. administration of 5 mg/kg TET, and the LPS group were i.v. administrated with 5 mg/kg TET 6-h after i.t. injection of 50 μL LPS (5 mg/mL) to produce acute pneumonia. The whole blood samples were collected at 0.033, 0.083, 0.25, 0.5, 1, 2, 4, 8, 24, 48, 72, 96, 120, and 144 h after TET administration and stored at −40 °C for LC-MS/MS analysis. Major PK parameters were calculated from WinNonlin 7.0. * *p* < 0.05 (compared to the normal mice group); ** *p* < 0.01 (compared to the normal mice group).

**Table 2 ijms-23-12948-t002:** Pharmacokinetic parameters of TET in major tissues and blood after a single i.t. administration in normal and LPS-treated mice (*n* = 3, Mean ± SD).

Parameters	Unit	Spleen	Kidney	Liver	Lung	Intestine	Heart	Stomach	Adipose	Muscle	Brain	Blood
Normal mice												
t_1/2_	h	83.8 ± 22.7	63.4 ± 8.09	47.5 ± 5.47	65.1 ± 18.3	44.7 ± 1.94	96.8 ± 6.46	64.0 ± 16.5	54.0 ± 21.5	49.2 ± 4.06	51.6 ± 17.2	83.2 ± 15.4
T_max_	h	12.00 ± 10.6	3.33 ± 1.15	4.00 ± 0	13.3 ± 9.24	4.00 ± 0	2.67 ± 1.15	4.67 ± 3.06	5.33 ± 2.31	4.00 ± 3.16	2.00 ± 0	3.33 ± 1.15
C_max_	μg/mL	89.7 ± 7.09	107 ± 16.6	83.3 ± 9.29	60.3 ± 3.21	46.7 ± 25.9	14.2 ± 3.16	22.3 ± 12.4	7.38 ± 1.79	6.12 ± 0.44	10.4 ± 9.53	2.67 ± 0.05
AUC_(0-t)_	h·μg/mL	7943 ± 665	6514 ± 691	4863 ± 334	4307 ± 1003	1459 ± 134	880 ± 22.2	929 ± 163	421 ± 94.4	359 ± 79.8	168 ± 60.6	175 ± 12.3
AUC_(0-∞)_	h·μg/mL	11,282 ± 629	8159 ± 422	5485 ± 237	5295 ± 721	1632 ± 130	1407 ± 21.5	1121 ± 207	501 ± 88.3	410 ± 80.1	190 ± 51.4	249 ± 16.4
MRT	h	112 ± 34.7	80.1 ± 16.6	53.9 ± 8.29	78.3 ± 24.5	50.5 ± 2.31	137 ± 0.09	70.0 ± 14.0	67.7 ± 37.1	57.7 ± 7.81	56.8 ± 26.5	111 ± 23.1
K_p_	/	45.7 ± 6.62	37.5 ± 5.70	28.0 ± 3.49	24.7 ± 6.06	8.35 ± 0.67	5.05 ± 0.38	5.29 ± 0.66	2.39 ± 0.40	2.04 ± 0.38	0.97 ± 0.39	1.00
LPS-treated mice												
t_1/2_	h	111 ± 18.3	77.8 ± 18.1	69.3 ± 14.8	/	41.7 ± 4.93	449 ± 259	61.4 ± 13.0	75.2 ± 35.2	78.8 ± 31.3	92.6 ± 27.2	95.5 ± 17.6
T_max_	h	17.3 ± 11.5	5.33 ± 2.31	3.33 ± 1.15	144 ± 0.00 **	2.00 ± 0.00 **	4.00 ± 0.00	3.33 ± 1.15	18.7 ± 9.24	6.67 ± 2.31	5.33 ± 2.31	24.0 ± 0.00 **
C_max_	μg/mL	89.7 ± 10.7	112 ± 11.5	76.3 ± 7.02	96.2 ± 15.3 *	76.5 ± 10.9	10.8 ± 4.01	27.2 ± 9.32	5.92 ± 1.41	7.51 ± 2.11	6.43 ± 3.79	1.94 ± 0.225 **
AUC_(0-t)_	h·μg/mL	9425 ± 942	7939 ± 1171	5889 ± 178 **	10,004 ± 1109 **	2390 ± 196 **	940 ± 381	1340 ± 192 *	566 ± 163	483 ± 89.5	129 ± 46.2	219 ± 17.7
AUC_(0-∞)_	h·μg/mL	15,987 ± 1318 **	11,010 ± 3084	7427 ± 632 **	/	2591 ± 271 **	4089 ± 3400	1645 ± 323	816 ± 429	674 ± 240	179 ± 38.4	326 ± 12.9 *
MRT	h	157 ± 29.5	101 ± 30.0	82.1 ± 15.6	/	42.4 ± 6.38	643 ± 376	73.3 ± 17.7	102 ± 57.9	101 ± 44.4	112 ± 35.4	133 ± 26.1
K_p_	/	43.3 ± 6.88	36.2 ± 3.83	27.0 ± 1.72	45.8 ± 6.49 *	11.0 ± 1.35 *	4.27 ± 1.59	6.13 ± 0.87	2.57 ± 0.62	2.20 ± 0.31	0.59 ± 0.17	1.00

Thirty BALB/c mice were randomly divided into ten groups (*n* = 3). Five groups of mice were intratracheally injected with 15 mg/kg TET (dissolved in 22.5 mg/mL aspartic acid solution) as control groups, and the other five groups received i.t. administration of 15 mg/kg TET 6-h after i.t. injection of 50 μL LPS (5 mg/mL) as model groups. One control group and one model group were sacrificed simultaneously at the set time points of 2, 4, 8, 24, and 144 h after the administration of TET. After a cardiac puncture to obtain blood samples, tissue samples including heart, liver, spleen, lung, kidney, stomach, intestine, and brain were immediately harvested and all the tissues were thoroughly rinsed in ice physiological saline. All the tissue samples were weighed and stored with blood samples at −40 °C for LC-MS/MS analysis. Major PK parameters were calculated from WinNonlin 7.0. Tissue partition ratio (K_p_) of each tissue was obtained from AUC in tissue divided by AUC in blood. * *p* < 0.05 (compared to the normal mice group); ** *p* < 0.01 (compared to the normal mice group).

**Table 3 ijms-23-12948-t003:** The metabolic stability of TET in normal and LPS-induced ALI mice liver microsomes (*n* = 3, Mean ± SD) correlated with in vivo observed clearance (*n* = 6, Mean ± SD).

Groups	In Vitro			In Vivo
	t_1/2_ (min)	CL_int,scaled_ (mL/min/mg Protein)	CL_hep,blood_ (mL/h/kg)	CL_blood_ (mL/h/kg)
Normal mice	52.8 ± 1.75	103 ± 3.40	140 ± 4.50	172 ± 22.1
LPS-treated mice	84.0 ± 0.780 **	65.0 ± 0.603 **	89.0 ± 0.814 **	80.9 ± 22.3 **

** *p* < 0.01 (compared to the normal mice group). The metabolic stability of TET (1.0 µM) was examined in NADPH-supplemented mice liver microsomes (0.5 mg/mL protein concentration) for 60 min at 37 °C. In vitro stability data governed by the half-life (t_1/2_) were scaled to apparent intrinsic clearance (CL_int,scaled_) and hepatic blood clearance (CL_hep,blood_) using the well-stirred model described in the Materials and Methods Section. Where, f_up_, f_u,mic_, and R_b/p_ were 0.01, 0.18, and 2.39 obtained from in vitro experiments. In vivo CL_blood_ obtained from the in vivo clearance after i.v. administration.

**Table 4 ijms-23-12948-t004:** The uptake parameters of TET by OATP-transfected and MOCK HEK293 cells (*n* = 3, Mean ± SD).

OATP	TET			Positive Control		
	Uptake Clearance (nM/min/mg Protein)	Uptake Ratio	Inhibition Ratio	Uptake Clearance (nM/min/mg Protein)	Uptake Ratio	Inhibition Ratio
MOCK	131 ± 11.3	1.21	11.1	2.4 ± 0.03 ^#^	18.4	97.7
OATP1B1	158 ± 35.1			44.2 ± 1.65 **^, ##^		
MOCK+RIF	117 ± 12.8	1.2		2.63 ± 0.08	1.36	
OATP1B1+RIF	141 ± 13.7			3.57± 0.03 **		
MOCK	126 ± 10.9	1.2	20.6	2.7 ± 0.07	11.1	96.6
OATP1B3	151 ± 25.5			30.0 ± 1.46 **^, ##^		
MOCK+RIF	119 ± 8.03	1.01		2.73 ± 0.08	1.34	
OATP1B3+RIF	120 ± 8.61			3.66 ± 0.03 **		

In the experiments using transporter expression systems, the uptake of 17β-estradiol-17β-D-glucuronide (reference compound for OATP1B1 and OATP1B3) for 10 min, and TET for 5 min were simultaneously determined, respectively. OATP1B1- and OATP1B3-mediated transport was calculated by subtracting the uptake in mock cells from that in OATP1B1- and OATP1B3-expressing cells. The method is described in detail in the Materials and Methods Section. ** *p* < 0.01 (compared to the MOCK group); ^#^
*p* < 0.05, ^##^
*p* < 0.01 (compared to the RIF group).

**Table 5 ijms-23-12948-t005:** The bidirectional transport of TET in MOCK and MDR1-transfected LLC-PK1 cells (*n* = 3, Mean ± SD).

Mock Cell	P_app_		ER	MDR1-Transfected	P_app_		ER
	A-B	B-A			A-B	B-A	
Atenolol	0.00	0.12 ± 0.01	\	Atenolol	1.23 ± 0.08 **	0.12 ± 0.01 **	0.84
Metoprolol	32.22 ± 2.80	24.99 ± 0.30	0.7	Metoprolol	28.23 ± 0.65	28.32 ± 2.07	1.00
Digoxin	2.25 ± 0.26	1.74 ± 0.20	0.77	Digoxin	0.98 ± 0.11 **^,##^	11.27 ± 0.77 **^,##^	11.54
Digoxin + TAR	2.08 ± 0.09	1.66 ± 0.21	0.80	Digoxin + TAR	5.14 ± 1.02 **	5.31 ± 0.85 **	1.03
TET	1.22 ± 0.07 ^##^	3.83 ± 0.12 ^#^	3.13	TET	0.52 ± 0.19 **	14.02 ± 2.95 **^,#^	32.46
TET + TAR	1.88 ± 0.14	5.21 ± 0.73	2.17	TET + TAR	0.37 ± 0.05 *	6.03 ± 1.50	16.55

Transport experiments in either the apical-to-basolateral direction (A-B) or the basolateral-to-apical direction (B-A) were conducted in MDR1 expressing LLC-PK1 cells or MOCK LLC-PK1 cells. Donor solution (HBSS with 1% BSA) containing 2 μM TET or cocktailed positive control drugs (10 μM metoprolol for high permeability, 10 μM atenolol for low permeability, and 2 μM digoxin for *p*-gp substrate) was added into the apical or basolateral compartment and a sample (50 μL) was immediately taken as C_0_. Tariquidar (TAR, 5 μM) as a selective MDR1 inhibitor co-incubated with TET or positive control compounds. After incubating the plates at 37 °C for 2 h, samples from the receiving and donor wells (50 μL) were taken simultaneously. All the samples were stored at −40 °C for LC-MS/MS analysis. P_app(AB)_ and P_app(BA)_ and ER were calculated according to Equations (8) and (9) in the Materials and Methods Section. * *p* < 0.05, ** *p* < 0.01 (compared to the MOCK group); ^#^
*p* < 0.05, ^##^
*p* < 0.01 (compared to the TAR group).

## Data Availability

The authors confirm that the data supporting the findings of this study are available within the article.

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
