# Peer review of "Alternation of Organ-Specific Exposure in LPS-Induced Pneumonia Mice after the Inhalation of Tetrandrine Is Governed by Metabolizing Enzyme Suppression and Lysosomal Trapping"

_ijms, 2022, doi:10.3390/ijms232112948_

Round 1

Reviewer 1 Report

This study examines the biological activities of TET in therapeutic use. Positive that the in vitro and in vivo results are in agreement. The conclusions reached by the authors are interesting: downregulation of CYP3A activity and bidirectional transport.
In particular, some corrections would be necessary:
- Zoom in on some figures, now not very legible, such as 2
- report what statistical analysis was carried out
- check that all abbreviations are explicit (for example ALI)
- briefly describe what CYP3A is

Author Response

Dear editor and dear reviewers:

Re: Manuscript ID: ijms-1971284 and Title: CYP3A and lysosomal trapping impact the biodistribution of inhaled tetrandrine in inflamed mice.

We really appreciate your letter and the reviewers’ decision and constructive comments related to our manuscript entitled “CYP3A and lysosomal trapping impact the biodistribution of inhaled tetrandrine in inflamed mice.” (ID: ijms-1971284). Those comments are all valuable and very helpful for revising and improving our paper as the important guiding significance to our research. To address your and the reviewers’ concerns, we have provided relevant information and improved the reorganized manuscript. The point-by-point responses are shown below in blue underlined. The revised parts in the manuscript are marked and uploaded together with a clean version. We would like to thank you and the referees again for taking the time to review our manuscript. 

Comments from Reviewer #1:

This study examines the biological activities of TET in therapeutic use. Positive that the in vitro and in vivo results are in agreement. The conclusions reached by the authors are interesting: downregulation of CYP3A activity and bidirectional transport.
In particular, some corrections would be necessary:
1. Zoom in on some figures, now not very legible, such as 2

Thank you for your advice. We modified the resolution of the figures and uploaded each figure separately to avoid loss of pixels by compression.

  1. report what statistical analysis was carried out

Thank you for your advice. We added the statistical treatment of data following your suggestion. Please see 4.8. Data processing and statistical analysis (line #537-540).

  1. check that all abbreviations are explicit (for example ALI)

Thank you for your advice. We checked all the abbreviations throughout the paper following your suggestion.

  1. briefly describe what CYP3A is

Thank you for your advice. CYP3A is the most abundant, clinically significant group of cytochrome P-450 isoenzymes. The CYP3A group is composed of three major isoenzymes: CYP3A4, CYP3A5, and CYP3A7. Since these enzymes are so closely related (having as much as 97% sequence homology), they often are referred to collectively by the subfamily name, CYP3A. We added the full name of CYP3A (cytochrome P-450 3A) and its important contribution to drug metabolism in the discussion (line # 285-286).

Reviewer 2 Report

Wang et al in their manuscript entitled as "Alteration of organ-specific exposure in LPS-induced mice after inhaled tetrandrine is governed by metabolizing enzymes suppression and lysosomal trapping" provides a interesting results of the pharmacokinetics profile of traditional Chinese medicine Tetrandrine (TET) using in vitro and in vivo models of acute lung injury. The manuscript provides a novel and useful information about mechanistic action of TET at tissue and molecular levels which paves the way for its further clinical development. Overall, the manuscript content is insightful but still it requires substantial modification especially the presentation of data.

Major Comments

1) All the figures need to be upgraded for high resolution and size. In the current format they are not readable when printed. Upon zoom they appear pixelated. 

2) The labels of the figures appear smaller than regular size.

3) Authors showed statistics on table 1 but statistics for table 2, 4 and 5 are missing.

4) Why there is no quantitation graph for TET (in mouse plasma) using LCMS/MS analysis. Graphical data using reference standard will be helpful to understand the PK properties of TET.

5) On page 6 of 17, Line 483 to 494 has a different font size which makes it inconsistence with manuscript text. This should be corrected.

6) Regarding CYP3A data, why authors did not study or show data of other CYP enzymes that may have affected by TET exposure? No rationale was found or discussed in the manuscript.

7) If TET metabolism is described as pH driven lysosomal trapping then why the chemokines involved in pH regulation and impacted by TET were not studied or discussed? 

8) There is an inconsistency throughout the text regarding use of italics to quote in vitro and in vivo wordings. Should be corrected.

Author Response

Dear editor and dear reviewers:

Re: Manuscript ID: ijms-1971284 and Title: CYP3A and lysosomal trapping impact the biodistribution of inhaled tetrandrine in inflamed mice.

We really appreciate your letter and the reviewers’ decision and constructive comments related to our manuscript entitled “CYP3A and lysosomal trapping impact the biodistribution of inhaled tetrandrine in inflamed mice.” (ID: ijms-1971284). Those comments are all valuable and very helpful for revising and improving our paper as the important guiding significance to our research. To address your and the reviewers’ concerns, we have provided relevant information and improved the reorganized manuscript. The point-by-point responses are shown below in blue underlined. The revised parts in the manuscript are marked and uploaded together with a clean version. We would like to thank you and the referees again for taking the time to review our manuscript. 

Comments from Reviewer #2:

Wang et al in their manuscript entitled as "Alteration of organ-specific exposure in LPS-induced mice after inhaled tetrandrine is governed by metabolizing enzymes suppression and lysosomal trapping" provides a interesting results of the pharmacokinetics profile of traditional Chinese medicine Tetrandrine (TET) using in vitro and in vivo models of acute lung injury. The manuscript provides a novel and useful information about mechanistic action of TET at tissue and molecular levels which paves the way for its further clinical development. Overall, the manuscript content is insightful but still it requires substantial modification especially the presentation of data.

Thank you for your positive comment.

  1. All the figures need to be upgraded for high resolution and size. In the current format they are not readable when printed. Upon zoom they appear pixelated. 

Thank you for your advice. We modified the resolution of the figures and uploaded each figure separately to ensure the quality,

  1. The labels of the figures appear smaller than regular size.

Thank you for your advice. We adjusted the font size for consistency following your suggestion.

  1. Authors showed statistics on table 1 but statistics for table 2, 4 and 5 are missing.

Thank you for your advice. We reported the statistical analysis results for table 2, 4, and 5 in the revision.

  1. Why there is no quantitation graph for TET (in mouse plasma) using LCMS/MS analysis. Graphical data using reference standard will be helpful to understand the PK properties of TET.

Thank you for your advice. We added the blood PK curves of TET in figure 2 to visualize the PK behavior of TET in the circulation system directly.

  1. On page 6 of 17, Line 483 to 494 has a different font size which makes it inconsistence with manuscript text. This should be corrected.

Thank you for your advice. Corrections have been made.

  1. Regarding CYP3A data, why authors did not study or show data of other CYP enzymes that may have affected by TET exposure? No rationale was found or discussed in the manuscript.

Thank you for your question. The CYP reaction phenotyping of TET is mainly CYP3A with a fraction of metabolism (fm) value of more than 0.9 in our previous study. We illustrated the results in the article. Please see line #286-287.

  1. If TET metabolism is described as pH driven lysosomal trapping then why the chemokines involved in pH regulation and impacted by TET were not studied or discussed? 

Thank you for your question. It was because the application of monensin significantly reduced the accumulation of TET in cells (Figure 5B), and because monensin has been testified to have the ability to transport protons across cellular lipid membranes to raise the lysosomal pH and reduce the difference in pH between lysosomes and the surrounding cytosol [37], that we deduced that the intracellular accumulation of TET was driven by pH-dependent lysosomal trapping. Many weakly alkaline drugs are lysosomotropic compounds. When a cell is exposed simultaneously to two different lysosomotropic compounds they compete for intra-lysosomal protonation and their relative lysosomal accumulation depends on their relative concentrations and respective pKa values. We made further discussion in the discussion part to address your concern (line #308-312).

  1. There is an inconsistency throughout the text regarding use of italics to quote in vitro and in vivo wordings. Should be corrected.

Thank you for your advice. Corrections have been made.